# Estimation and Spatio-Temporal Change Analysis of NPP in Subtropical Forests: A Case Study of Shaoguan, Guangdong, China

**Tao Li, Mingyang Li \*, Fang Ren and Lei Tian** 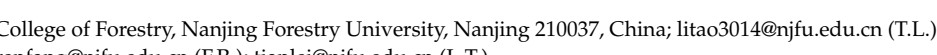

College of Forestry, Nanjing Forestry University, Nanjing 210037, China; litao3014@njfu.edu.cn (T.L.); renfang@njfu.edu.cn (F.R.); tianlei@njfu.edu.cn (L.T.)
\* Correspondence: lmy196727@njfu.edu.cn; Tel.: +86-025-8542-7327

**Abstract:** Exploring the spatial and temporal dynamic characteristics of regional forest net primary productivity (NPP) in the context of global climate change can not only provide a theoretical basis for terrestrial carbon cycle studies, but also provide data support for medium- and long-term sustainable management planning of regional forests. In this study, we took Shaoguan City, Guangdong Province, China as the study area, and used Landsat images and National Forest Continuous Inventory (NFCI) data in the corresponding years as the main data sources. Random forest (RF), multiple linear regression (MLR), and BP neural network were the three models applied to estimate forest NPP in the study area. Theil–Sen estimation, Mann–Kendall trend analysis and the standard deviation ellipse (SDE) were chosen to analyze the spatial and temporal dynamic characteristics of NPP, whereas structural equation modeling (SEM) was used to analyze the driving factors of NPP changes. The results show that the performance of the RF model is better than the MLR and BP neural network models. The NPP in the study area showed an increasing trend, as the NPP was 5.66 t·hm$^{-2}$·a$^{-1}$, 7.68 t·hm$^{-2}$·a$^{-1}$, 8.17 t·hm$^{-2}$·a$^{-1}$, 8.25 t·hm$^{-2}$·a$^{-1}$, and 10.52 t·hm$^{-2}$·a$^{-1}$ in 1997, 2002, 2007, 2012, and 2017, respectively. Spatial aggregation of NPP was increased in the period of 1997–2017, and the center shifted from the mid-west to the southwest. In addition, the forest stand factors had the greatest effect on NPP in the study area. The forest stand factors and environmental factors had a positive effect on NPP, and understory factors had a negative effect. Overall, although forest NPP has fluctuated due to the changes of forestry policies and human activities, forest NPP in Shaoguan has been increasing. In the future, the growth potential of NPP in Shaoguan City can be further increased by continuously expanding the area proportion of mixed forests and rationalizing the forest age group structure.

**Keywords:** net primary productivity; remote sensing inversion; dynamic change; driving factors; Shaoguan City

## 1. Introduction

With the intensification of global climate change, the global carbon cycle has become one of the core issues in global climate change research [1–3]. As the main body of the terrestrial ecosystem and the largest carbon reservoir in the terrestrial ecosystem, the forest ecosystem fixes about two-thirds of the carbon in the whole terrestrial system every year, and its role in regulating global carbon balance, mitigating the rising concentration of greenhouse gases such as $CO_2$ in the atmosphere, and maintaining global climate is irreplaceable [4–6]. The net primary productivity (NPP) of forests is the amount of organic matter accumulated per unit area and per unit time, which is expressed as the portion of organic carbon fixed by photosynthesis minus the portion consumed by plants themselves through respiration [7]. Estimation of NPP is the basis for the study of the functioning of matter and energy in ecosystems, reflects the production capacity of vegetation communities under natural environmental conditions, and also directly affects the carbon stocks of above-ground plant parts, below-ground root parts, soil carbon pools, and the carbon

interception of the whole forest ecology. As a component of the surface cycle, NPP can directly reflect the production capacity of forests under natural environmental conditions, as it is one of the important indicators for evaluating the sustainable development of the forest ecosystem and is the main indicator for judging the carbon sink of ecosystems and regulating ecological processes [8]. In the context of intensifying global climate change and human activities, considering the important role played by NPP in the global carbon cycle, carbon sequestration, carbon storage, and global change, the research on the NPP estimation of forest vegetation is carried out to provide a scientific basis for the making of a long-term sustainable forest management plan at the regional scale.

At the spatial scale, the estimation of forest NPP is divided into sample plot observations, regional simulation, and global simulation. The localization observation uses the extrapolation method, which is the extrapolation of sample survey points to the whole region. Although the rationality is insufficient, the estimation uses spatially measured data; thus, it can still be used as a reference for NPP estimation [9]. At the regional or global scale, because forest ecosystems are the most complex types of structural levels and functional behaviors on Earth, the impacts of environmental changes and human activities on forests and the feedback effects of forests are long-term. However, the availability of productivity observation data points on forest areas worldwide is extremely limited; therefore, the model estimation of productivity becomes an important research method [10,11]. Models for estimating NPP can be classified into four categories: the statistical model, parametric model, process model, and ecology and remote sensing coupling model. Statistical models include the Miami model, the Thornthwaite memorial model, etc. A large number of empirical studies have shown that the Thornthwaite memorial model is sensitive only to changes in precipitation. The Miami model, though sensitive to both precipitation and temperature, is more sensitive to temperature [12]; in addition, this model does not take into account the influence of structural changes of vegetation itself, nor the influence of atmospheric conditions and site conditions, and only uses mathematical methods to derive the relationship between NPP and major climatic factors. Therefore, the estimation results obtained from the model are approximate. Parametric models mainly include the CASA model, GLO-PEM model, VPM model, etc., among which the CASA model has become a mature model for estimating NPP. Potter et al. [13] used the CASA model to estimate global biomass and productivity in 1980. Wen [14] used the CASA model to estimate NPP as a vegetation growth indicator and combined it with other climate factors to study the global vegetation's response to climate warming from 1982 to 2013, but there are uncertainties in the simulation results for vegetation NPP because the parametric models cannot explain the mechanisms of productivity changes in terms of physiological ecology. The process model takes into account the physiological characteristics of the vegetation and environmental factors to simulate the vegetation growth and development process, but the model is complex and requires more parameters, which are the two main defects of the process model. The ecology and remote sensing coupling model combines the advantages of the plant ecophysiological process model and remote sensing parameter model, which can make a global-scale NPP estimation, and the model parameters can be obtained using remote sensing technology. However, this kind of model is more complex and requires more parameters, and subjective factors have a greater influence on the investigator and parameter determination.

From remote sensing data, information on forest cover status and forest spectral characteristics can be obtained at a large regional scale, although this information is closely related to forest productivity. Therefore, optical sensors and active sensors have been widely used in forest net primary productivity estimation [15]. Since the spatial resolution of Landsat time series stacks (LTSS) image elements is 30 m $\times$ 30 m and the image element size is close to the area of the forest management unit, it makes up for the shortage of single-period images in monitoring long-term forest dynamics, and is therefore often used for regional forest productivity estimation. Remote sensing estimation combined

with ground survey data has become one of the important tools in regional scale forest productivity estimation.

Forest productivity is related to age and vegetation growth status, whereas the remote sensing characteristic variables extracted by traditional methods do not contain such information. Forest canopy density is the ratio of the forest vegetation to projected area observed from the air, and is numerically the same value as crown density as an indicator of stand density. It is one of the important components of stand structure, and it can be used to reflect the distribution of light, water, and other environmental factors entering the stand through the forest canopy. The magnitude of forest NPP is closely related to canopy density, and the accuracy of the model may be improved by introducing the canopy density variable into the forest NPP inversion model [16].

NPP varies spatially and temporally, and there are also many driving factors that cause spatial and temporal changes in NPP. Therefore, it is important to study the spatial and temporal dynamics of NPP and the driving factors for long-term sustainable forest management. The Theil–Sen median slope estimation and Mann–Kendall trend analysis are more robust to errors and can determine the significance of trends. Nyikadzino [17] used these methods to observe seasonal changes in precipitation in the Limpopo River Basin and found that rainfall in this area showed a non-significant decrease trend. Alhaji [18] used this method to study the temperature in Gombe State, Federal Republic of Nigeria and found that due to the effects of climate change, extreme weather caused a significant increase in the maximum and average temperature in the area. The SDE method takes into account three levels of center offset, directional offset, and angular offset, and can visually reflect the spatial variation results. Peng [19] studied the spatial and temporal distribution of PM2.5 concentrations in China from 1999 to 2011 using the SDE. In previous studies, the Theil–Sen median slope estimation, Mann–Kendall trend analysis, and SDE are mainly applied to the study of climate spatial and temporal variability over long time periods. At present, there are no relevant reports on the analysis of spatio-temporal dynamics and long-term change of forest NPP based on the Theil–Sen median slope estimation, Mann–Kendall trend analysis, and SEM analysis of driving factors. There are many drivers of spatial and temporal changes in forest NPP, which can be divided into two main categories: biotic natural factors and socio-economic factors. Wang [20] studied the changes in forest NPP and multi-level driving mechanisms in the Changbai Mountains, Northeast China, and the results showed that precipitation and vegetation cover were the key drivers.

Subtropical forests hold a special position in the global ecosystem and play an important role in the global terrestrial ecosystem material cycle and terrestrial carbon pool. China's subtropical forest is a unique forest ecosystem, characterized by rich forest types, a wide range of tree species, and high forest productivity [21]. Guangdong Province is located in the southeastern part of the Asian continent, south of the tropical sea, and is strongly influenced by the monsoon climate, with many typhoons and heavy rains in summer. Under this climate condition, the forest service functions of water conservation and soil conservation are particularly important. Since the influence of the Quaternary Ice Age is small, the flora in the province has had a long history of development, producing a rich variety of forest plant species and forming a flora with many ancient plants and relict plants [22,23]. Shaoguan City is located in the northern part of Guangdong Province and is an important part of the southern collective forest area. Due to the differences in topographic conditions, forest resource status, and economic development level, there is spatial and temporal heterogeneity of forest NPP and its drivers. Since the reform and opening up in 1978, the urbanization and industrialization process in Shaoguan City has been accelerated. The forests have been disturbed and damaged by human activities for a long time, the remnant native forest in the city has been decreasing, and the habitats for many wildlife have been deteriorating.

Until now, there are no relevant reports on the analysis of spatio-temporal dynamics and driving factors of forest NPP over a long time based on the Theil–Sen median slope estimation, Mann–Kendall trend analysis, SDE, and SEM. The main objectives of this study

are as follows: (1) using Landsat images and NFCI data as the main information sources to introduce the stand structure factor of forest canopy density, which is closely related to productivity, to estimate the NPP in Shaoguan City in 1997, 2002, 2007, 2012, and 2017, and to explore more accurate NPP estimation methods from the three RF, MLR, and BP neural network models; (2) to reveal the spatio-temporal change trend of forest NPP in a typical forest prefecture in a Chinese subtropical region; and (3) to identify driving factors of forest NPP dynamics to provide a scientific basis for making sustainable forest management plans.

## 2. Materials and Methods

### 2.1. Study Area

Shaoguan City (Figure 1) is located in the northern part of Guangdong Province (23°53′~25°31′N, 112°53′~114°45′E). The whole territory spans 186.30 km from east to west and 173.40 km from north to south. Shaoguan's topography is high in the north and low in the south, with the highest peak of Shikenggang (1902 m asl) in the north of Guangdong and the lowest point (35 m asl) in the south. Shaoguan belongs to the central subtropical humid monsoon climate zone and has a pleasant climate. The average annual temperature is 21 °C, with the temperature increasing from north to south in winter, and the temperature is almost the same in summer. Rainfall is abundant, with an average annual rainfall of 1700 mm. March–August is the rainy season, September–February is the dry season, and there is snow in the north in winter.

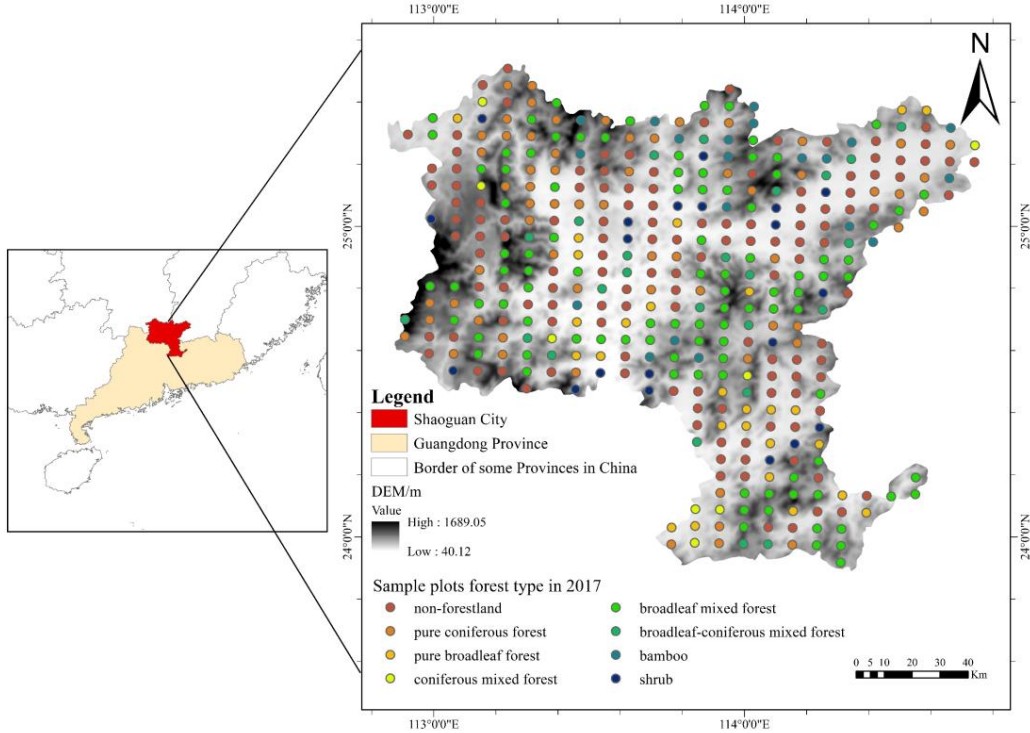

**Figure 1.** Location of Shaoguan City, Guangdong Province, together with the DEM.

Shaoguan is a national key forest area, being the important base of the timber forest, water source forest, and key moso bamboo in Guangdong Province, and is known as the biological gene pool of South China and the ecological shield of the Pearl River Delta. In 2021, the city had a forested area of 1,277,300 hm², with a forest coverage rate of 74.43%, forest greening rate of 74.90%, and stock volume of 96.52 million m³, ranking first in Guangdong Province, which is known as "The World's Most Complete Oasis Preserved on the Same Latitude as the Tropic of Cancer". The forest in Shaoguan is dominated by broadleaf mixed forests and broadleaf pure forests. The broad-leaved pure forests mainly include oak (*Quercus*), eucalyptus (*Eucalyptus robusta*), camphor tree (*Cinnamomum*

*camphora*), etc. These forests are followed by coniferous pure forests represented by horsetail pine (*Pinus massoniana*) and Chinese fir (*Cunninghamia lanceolata*), mixed coniferous forests, and bamboo forests represented by moso bamboo (*Phyllostachys heterocycle*) [24]. The specific information is shown in Table 1.

**Table 1.** Forest types in Shaoguan City.

| Main Forest Types | Standard of Division | Typical Tree Species | Characteristic |
|---|---|---|---|
| pure coniferous forest | stand volume of single coniferous species ≥65% | *Cunninghamia lanceolata* | fast growth, high volume per unit area |
| | | *Pinus massoniana* | wide distribution, main tree species for timber forest |
| pure broadleaf forest | stand volume of single broadleaf species ≥65% | *Eucalyptus robusta* | high proportion of young forests |
| | | *Acacia confusa* | higher volume per unit area, mainly planted forests |
| | | *Cinnamomum camphora* | grow faster, native hardwood species |
| broadleaf mixed forest | total stand volume of broadleaf species ≥65% | | few natural broad-leaved mixed forests, the dominant tree species is not obvious |
| broadleaf-coniferous mixed forest | total stand volume of coniferous or broadleaf species accounting for 35–65% | *Pinus massoniana-Schima superba* | tree growth is higher than their respective pure forests |
| coniferous mixed forest | total stand volume of coniferous species ≥65% | *Cunninghamia lanceolata-Pinus massoniana* | less pests and diseases |

Shaoguan City is rich in forest resources, and forest cover and forest stock volume are higher than the national average. Although Shaoguan City forest resources have had steady growth, due to frequent human activities, accelerating urbanization and industrialization, and irrational forestry policies, Shaoguan City forest resources still have certain problems, such as low forest quality and uneven forest age groups. Therefore, in this study, based on the Landsat series remote sensing images, we compare the estimation methods of forest NPP in the study area, explore the dynamic changes of forest NPP in long time series, and explore the driving factors affecting NPP.

*2.2. Data Acquisition and Preprocessing*

2.2.1. The Fixed Sample Data of National Forest Resources Continuous Inventory

The NFCI [25] takes provinces as the sampling population and adopts systematic sampling. According to the actual situation of each province, the sampling interval of each province is determined by the kilometer grid. Permanent sample plots are set up to conduct forest resource surveys. In Shaoguan City, there are 388 fixed sampling plots (25.82 × 25.82 m each) based on 6 km × 8 km spacing. The attributes of these plots include slope, slope direction, slope position, altitude, soil name, soil layer thickness, soil texture, humus thickness, average age, average diameter at breast height (DBH), average tree height, canopy density, tree species structure, live tree stock volume, and other investigation factors. The National Forestry and Grassland Administration is responsible for establishing the inventory plots and gathering data.

Before the estimation of NPP, the non-forestland sample plots with a stock volume of 0, such as water bodies and buildings, were removed from the NFCI. The NPP of forest consists of community growth (the sum of annual net growth of stems, branches, and roots of the tree layer and annual net growth of shrubs and the herbaceous layer) and annual withered volume. Based on the relationship between biomass and stock volume, and the function relationship between biomass, community growth, and annual withering, the biomass of different forest types was calculated based on the volume, the community

growth and annual withering were calculated based on the biomass, and the NPP of the forest was finally obtained for each sample plot. The specific calculation formula is shown in Table 2.

**Table 2.** Relationship between forest volume, biomass, and NPP of typical tree species (excerpt) [26].

| Forest Types | Relationship between Biomass and Volume | Relationship between Biomass and Community Growth | Relationship between Biomass and Annual Withering |
|---|---|---|---|
| coniferous and broadleaf mixed forest | B = V/(1.1731 + 0.0018 × V) | Y = B/(0.1038 × A + 0.0761 × B) | L = 3.46 |
| deciduous broadleaf forest | B = V/(0.6539 + 0.0038 × V) | Y = B/(0.2393 × A + 0.0495 × B) | L = B/(18.2460 + 0.0366 × B) |
| broadleaf mixed forest | B = V/(0.5788 + 0.0020 × V) | Y = B/(0.3018 × A + 0.0331 × B) | L = B/(9.1028 + 0.0575 × B) |
| cypress forest | B = V/(1.0202 + 0.0022 × V) | Y = B/(0.1132 × A + 0.0745 × B) | L = B/(9.8381 + 0.1337 × B) |
| fir forest | B = V/(1.2917 + 0.0022 × V) | Y = B/(0.4598 × A + 0.0069 × B) | L = B/(10.1320 + 0.0874 × B) |
| Pinus massoniana forest | B = V/(1.4254 − 0.0004 × V) | Y = B/(0.4046 × A + 0.0098 × B) | L = B/(15.4510 + 0.0225 × B) |
| other warm pine forest | B = V/(1.3624 − 0.0003 × V) | Y = B/(0.2423 × A + 0.0581 × B) | L = B/(18.9050 + 0.0422 × B) |
| evergreen broadleaf forest | B = V/(0.7883 + 0.0026 × V) | Y = B/(0.2503 × A + 0.0226 × B) | L = B/(20.5070 + 0.0383 × B) |
| deciduous broadleaf forest | B = V/(0.6539 + 0.0038 × V) | Y = B/(0.2393 × A + 0.0495 × B) | L = B/(18.2460 + 0.0366 × B) |

Note: B is biomass (t·hm$^{-2}$), V is volume (t·hm$^{-2}$), Y is the community growth(t·hm$^{-2}$), L is annual withering(t·hm$^{-2}$), A is average age (a).

2.2.2. Landsat Time Series Data

(1) Image pre-processing

Landsat images were pre-processed using ENVI 5.3 software. To eliminate the errors of the sensor, the images were first radiometrically calibrated [27], and then atmospheric correction [28] was performed using the FLAASH module of the ENVI software. The terrain in the study area is mainly mountainous and hilly, with large differences in elevation between the north and the south. The remote sensing images are influenced by the sensor orientation and the sun height and orientation, resulting in differences in brightness values due to the different illumination received by the shaded and sunny slopes [29]. The C-correction algorithm is used to correct the topography of the remote sensing images to eliminate the variation of radiance values caused by the topographic relief, so that the images can better reflect the spectral characteristics of the features [30]. Due to the failure of the Landsat-7 ETM+ on-board scan line corrector (SLC) after June 2003, data strips were lost in the images acquired after that date; thus, the 2007 and 2012 image data needed to be strip-repaired first and then pre-processed.

(2) Extraction of Feature Variables

According to previous studies [31], it is known that the productivity of forests is closely related to the conditions of forest stand, soil, topography, and climate. This study uses ENVI software to extract seven vegetation indices, including the normalized difference vegetation index (NDVI), ratio vegetation index (RVI), difference vegetation index (DVI), enhanced vegetation index (EVI), green vegetation index (GVI), perpendicular vegetation index (PVI) [32] and leaf area index (LAI) [33]. Tasseled cap transformation [34] was performed on the pre-processed Landsat remote sensing images, and the first three components of brightness, greenness, and wetness were chosen. Three window sizes of 3 × 3, 5 × 5, and 7 × 7 were selected when extracting texture features, and contrast, dissimilarity, mean, homogeneity, angular second moment, entropy, skewness, and correlation were calculated. Alternative independent variables of the NPP remote sensing estimation model are shown in Table 3.

**Table 3.** Alternative independent variables of NPP remote sensing estimation model.

| Variable Type | Variable Name | Code | Variable Type | Variable Name | Code |
|---|---|---|---|---|---|
| single band | blue band | B2 | stand structure | canopy density | FCD |
| | green band | B3 | tasseled cap | brightness index | Bri |
| | red band | B4 | | greenness index | Gre |
| | near-infrared band | B5 | | wetness index | Wet |
| | shortwave infrared band 1 | B6 | topographic factor | slope | Slope |
| | shortwave infrared band 2 | B7 | | elevation | DEM |
| texture feature | contrast | Bij_con | vegetation index | normalized difference vegetation index | NDVI |
| | dissimilarity | Bij_dis | | ratio vegetation index | RVI |
| | mean | Bij_mea | | difference vegetation index | DVI |
| | homogeneity | Bij_hom | | enhanced vegetation index | EVI |
| | angular second moment | Bij_asm | | green vegetation index | GVI |
| | entropy | Bij_ent | | perpendicular vegetation index | PVI |
| | skewness | Bij_ske | | leaf area index | LAI |
| | correlation | Bij_cor | | | |

Note: texture feature code of Bij_xxx: i is the band of 2–7, j is $3 \times 3$, $5 \times 5$, or $7 \times 7$ texture window size.

### 2.3. Research Method

#### 2.3.1. Calculation of Forest Canopy Density

According to JOSHI's [35] study, it is known that there are four methods for forest canopy density extraction: the artificial neural network (ANN), multiple linear regression technique (MLR), forest canopy density mapper (FCD), and maximum likelihood classification (MLC). Among the four methods, the average accuracy of forest canopy density obtained by the FCD model in three Southeast Asian countries was 92% [36]; thus, the FCD was chosen for this study. The FCD model is calculated using forest growth condition and is able to monitor temporal changes in forest canopy density. The FCD model is based on the Landsat remote sensing image extracted index and includes the advanced vegetation index (AVI), bare soil index (BI), and shadow index (SI). Compared with the NDVI, the AVI is more sensitive to the amount of vegetation, the BI increases with the increase of surface bareness, and the SI increases with the increase of forest density. Taking Landsat 8 OLI images as an example, the three indices and FCD were calculated as follows.

$$\text{AVI} = [(B_5 + 1)(65{,}536 - B_4)(B_5 - B_4)]^{\frac{1}{3}} \tag{1}$$

$$\text{BI} = \frac{(B_6 + B_4) - (B_5 + B_2)}{(B_6 + B_4) + (B_5 + B_2)} \times 100 + 100 \tag{2}$$

$$\text{SI} = [(65{,}536 - B_2)(65{,}536 - B_3)(65{,}536 - B_4)]^{\frac{1}{3}} \tag{3}$$

$$\text{FCD} = (VD \times SSI + 1)^{\frac{1}{2}} - 1 \tag{4}$$

where $B_2$–$B_6$ indicate the brightness values of blue, green, red, near-infrared, and shortwave infrared bands; AVI indicates the advanced vegetation index and AVI = 0 when $(B_5 - B_4)$ is less than 0; *VD* indicates the vegetation density value, which is synthesized from the vegetation index and shading index using a principal component analysis; and *SSI* indicates the scale shading index, which is calculated using the linear transformation function of the normalized shading index.

#### 2.3.2. Remote Sensing Estimation Model

A random forest (RF) is a compositional supervised learning algorithm that uses a bootstrap method to extract multiple samples from the original samples, establishes a decision tree for each resampled sample, and then combines the decision trees to obtain the

final prediction by voting [37]. A large number of theoretical and practical studies have proved that RF has a high prediction accuracy, good tolerance for outliers and noise, and can reduce the overfitting phenomenon because the samples of random forest-generated decision trees are randomly selected [38]. In this study, the RF model is executed by the randomForest function in the R language randomForest package. The number of decision trees (ntree) and the number of variables (mtry) to be extracted when splitting the decision trees need to be adjusted when using the model. The value of mtry is 1/3 of the number of independent variables when building the RF regression model, and the default value of mtry is 1 when the number of independent variables is less than 3 [39].

A multiple linear regression (MLR) has two or more independent variables. The multiple linear regression model uses the vegetation index, texture characteristics, original band, forest canopy density, and other factors as independent variables and the NPP of Shaoguan's fixed sample plots as dependent variables. The screening of independent variables is conducted using stepwise regression, of which the basic idea can be seen in Section 2.3.3 [40]. Stepwise regressions were performed using SPSS software, with stepwise criteria of F probabilities, and entry and deletion were set to be 0.05 and 0.1, respectively.

An artificial neural network (ANN) simulates neuronal activity with a mathematical model, and is an information processing system based on imitating the structure and function of neural networks in the brain. The basic idea of the back propagation (BP) neural network is that a learning process consists of forward propagation of the signal and backward propagation of the error. Forward propagation means that the input samples are processed in the input layer and then passed to the output layer after the hidden layer. If the actual output of the output layer does not match the expected output, it will be transferred to the back propagation of error, and the back propagation of error will back propagate the error of the output into the input layer in a certain form through the hidden layer, spread the error to all units in each layer, and then obtain the error signal of each unit in each layer, where the error will be used as the basis for correcting the cell weight. The process of adjusting the weights is the process of network learning and training until the network output error can be accepted and proceed to the preset learning times. This study used a three-layer structure of a BP neural network model, including the input layer, hidden layer, and output layer [41]. The BP neural network model in this study was constructed with the R language neuralnet function package, where the model is first built with the default number of nodes and hidden layers of the system, and the number of hidden layers is further increased according to the resultant error to improve the model accuracy.

### 2.3.3. Screening of Model Feature Variables

The RF defines two metrics to measure the importance of variables and that can be used to rank the variables: the first is %IncMSE, which is the percentage increase in prediction error per decision tree computed with out-of-bundle (OOB) data replacement; the second is IncNodePurity, which is the total reduction in node impurity when the decision tree nodes split, measured as the sum of squared residuals. Higher values of %IncMSE and IncNodePurity of the predictor variables indicate greater importance for model prediction [42]; thus, these two metrics were used to screen the variables added to the RF and BP neural network models, and this step was performed in the R software.

Stepwise regression is an important method for selecting the optimal explanatory variables for linear regression models and which mainly addresses the problem of how to select the explanatory variables when there are too many variables. Therefore, the explanatory variables selected for the regression model have a significant effect on the response variables [43]. The basic idea of stepwise regression is to introduce variables into the model one by one, perform an F-test after introducing each explanatory variable, and perform a t-test on the explanatory variables that have been selected one by one. When the explanatory variables initially introduced become no longer significant due to the introduction of later explanatory variables, they are removed to ensure that only significant variables are included in the regression equation before each new variable is introduced.

This is an iterative process that lasts until neither significant explanatory variables are selected into the regression equation nor insignificant explanatory variables are removed from the regression equation, to ensure that the final set of explanatory variables obtained is optimal [44]. However, in the process of stepwise regression, because there may be significant correlation between independent variables, it is easy to cause the problem of collinearity between model variables. In order to eliminate the influence of collinearity between variables on the model, the variance inflation factor (VIF) between variables is calculated to test whether there is collinearity between variables. According to the study, the VIF is usually larger than 7.5, which indicates that serious collinearity between the variables exists and it needs to be eliminated [45,46]. The final variable screening of the multiple linear regression model in this study was performed using stepwise regression, which was executed in SPSS software with the confidence level set at 95%, and the F-test probabilities of entry and deletion of predictor variables were set at 0.05 and 0.10, respectively.

The specific screening results are shown in Table 4.

**Table 4.** The selected predictor variables for LF, BP neural network, and MLR models.

| Year | Predictor Variable of LF and BP | Predictor Variable of MLR |
|---|---|---|
| 1997 | DEM, B37_con, B27_con, B23_ske, B23_ent, B23_asm, B23_mea, B35_cor, Slope, B73_ent, B73_mea, B45_cor, B33_ent, B43_mea, B73_asm, B73_ske, B33_mea | DEM, B73_ske, B27_con |
| 2002 | B67_mea, FCD, B33_con, B47_con, B35_con, B25_hom, B63_dis, Slope, B55_mea, NDVI, B25_ske, B43_dis, B35_dis, B45_con, B27_ske, B77_con, DEM, B37_hom, B37_con, B23_hom, B57_mea, B27_con, B65_asm, B65_hom, B63_hom, B67_con | B37_con, B33_hom |
| 2007 | B63_dis, Wet, B25_con, B27_con, B45_con, B65_dis, B23_con, DEM, B2, B33_con, RVI, Bri, B4, B75_dis, FCD | RVI, B35_cor, B77_ent |
| 2012 | Wet, B2, DEM, B65_con, B47_ent, B73_ent, Slope, RVI, B73_mea, FCD, B27_con, B73_con, B4, B23_con, LAI, B45_mea, B3, B47_mea, B25_con, B53_hom | B25_con, B47_ske, B7, B55_dis, B45_ske |
| 2017 | B73_cor, B45_asm, B75_cor, B33_ske, B23_con, DEM, B53_hom, FCD, B2, B63_cor, B53_asm, RVI, B75_ent, B55_asm, NDVI, B45_con | B35_con, B75_cor, DEM, B27_con, B55_hom, B53_asm, Slope |

Note: DEM is elevation, Slope is slope, FCD is canopy density, NDVI is difference normalized vegetation index, RVI is ratio vegetation index, LAI is leaf area index, Wet is wetness index, Bri is brightness index, B2 is blue light band, B3 is green light band, B4 is red light band. In Bij_xxx, i is band, where 2–7 are blue light band, green light band, infrared band, shortwave infrared band 1, and shortwave infrared band 2, respectively; j is texture window size, where 3, 5, and 7 denote 3 × 3, 5 × 5, and 7 × 7 windows, respectively; con is contrast, dis is dissimilarity, mea is mean, hom is homogeneity, asm is angular second moment, ent is entropy, ske is skewness, and cor is correlation.

## 2.3.4. Model Accuracy Evaluation

After the model is established, it is necessary to check the goodness of fit and applicability of the model, to analyze the advantages and disadvantages of the model, and finally, to choose the optimal model.

In this study, we used 10-fold cross-validation to verify the accuracy of the model [47]. The method divided the data into 10 parts, where 9 of them are used as training data and 1 as test data in turn, and the mean value of 10 times was used as an estimate of the accuracy of the model.

There are many indicators for evaluating estimation models, such as coefficient of determination ($R^2$), root mean square error (RMSE), mean absolute error (MAE), etc. [48]. These metrics are usually used to determine the strength of the model by performing a comparative analysis between predicted and measured values. In this study, $R^2$, RMSE, and MAE are used to evaluate the accuracy of the model. $R^2$ reflects the proportion of the total variation of the dependent variable that can be explained by the independent variable through the regression relationship, and its value interval is usually between (0, 1). MAE is the mean value of the absolute error, which can better reflect the actual situation of the prediction error, and is equal to 0 when the predicted value and the true value match exactly; the larger the error, the larger the numerical value.

In this study, the model prediction performance is mainly evaluated by calculating the $R^2$, RMSE, and MAE of the model for accuracy evaluation.

$$R^2 = 1 - \frac{\sum_{i=1}^{n}(y_i - \hat{y}_i)^2}{\sum_{i=1}^{n}(y_i - \overline{y})^2} \tag{5}$$

$$RMSE = \sqrt{\frac{\sum_{i=1}^{n}(y_i - \hat{y}_i)^2}{n-1}} \tag{6}$$

$$MAE = \frac{1}{n}\sum_{i=1}^{n}|\hat{y}_i - y_i| \tag{7}$$

In the formula, $y_i$ is the actual observed value, $\hat{y}_i$ is the model prediction, $\overline{y}$ is the average of the actual values, and $n$ is the sample size.

### 2.3.5. Theil–Sen Median Slope Estimation and Mann–Kendall Trend Analysis

The Theil–Sen median slope estimation (also known as Sen slope estimation) is a robust, nonparametric statistical trend calculation method which can reduce data outliers, and combined with the Mann–Kendall trend analysis (MK test) method is suitable for trend analysis of long time series data [49]. This method does not require the data to obey normal distribution, has a strong resistance to data errors with a more solid statistical theoretical basis for the test of significance level, and the results are more scientific and reliable.

The Sen slope estimate is calculated as

$$\beta = Median\left(\frac{NPP_j - NPP_i}{j - i}\right), \quad 1997 \leq i \leq j \leq 2017 \tag{8}$$

In the formula: *Median ()* represents the median value; when $\beta > 0$, it indicates that the forest NPP shows an upward trend; when $\beta = 0$, it indicates that the forest NPP has no change; when $\beta < 0$, it indicates that the forest NPP shows a downward trend. The specific grading is shown in Table 5.

**Table 5.** Trend grading of MK test [50].

| $\beta$ | Z | Trend Grading |
|---|---|---|
| $\beta > 0$ | 2.58 < Z | extremely significant increase |
| | 1.96 < Z ≤ 2.58 | significant increase |
| | 1.65 < Z ≤ 1.96 | least-significant increase |
| | Z ≤ 1.65 | non-significant increase |
| $\beta = 0$ | Z | no change |
| $\beta < 0$ | Z ≤ 1.65 | non-significant decrease |
| | 1.65 < Z ≤ 1.96 | least-significant decrease |
| | 1.96 < Z ≤ 2.58 | significant decrease |
| | 2.58 > Z | extremely significant decrease |

The MK test is a non-parametric statistical test for trend of time series, used to judge the significance of the trend. The data of time series do not need to obey the normal distribution, independent of a few outliers and missing values. The MK test is more applicable to non-normally distributed data, and is usually used to explain the change in the trend of the time series of forest NPP. In the trend test, the original hypothesis $H_0$ indicates that no trend exists in data set $x$; the opposing hypothesis $H_1$ indicates that there is a monotonic trend in data set $x$.

Suppose $x_1, x_2, \ldots, x_n$ are time series variables and the constructed statistic is

$$S = \sum_{j=1}^{n-1}\sum_{i=j+1}^{n} sgn(x_i - x_j) \tag{9}$$

$$sgn\left(x_i - x_j\right) = \begin{cases} 1, & x_i - x_j > 0 \\ 0, & x_i - x_j = 0 \\ -1, & x_i - x_j < 0 \end{cases} \tag{10}$$

In the formula: $x_i$ and $x_j$ are the corresponding data (NPP) values of the $i$-th and $j$-th years, respectively, and $i > j$; $n$ is the length of the data set. Then, there is

$$Z = \begin{cases} \frac{S-1}{\sqrt{Var(S)}}, & S > 0 \\ 0, & S = 0 \\ \frac{S+1}{\sqrt{Var(S)}}, & S < 0 \end{cases} \tag{11}$$

In the formula: $Z$ is a normally distributed statistic; $Var(S)$ is the variance. The original hypothesis is rejected if $|Z| \geq Z_{(\alpha/2)}$ at a given $\alpha$ significant level, i.e., there is a significant upward or downward trend in the time series data at the $\alpha$ significant level.

### 2.3.6. Standard Deviational Ellipse

The standard deviational ellipse [51–53] is an analysis method to characterize the spatial distribution characteristics, including the center coordinate, the rotation angle, and the standard deviation along the long axis (i.e., $y$-axis) and the short axis (i.e., $x$-axis). These elements, respectively, represent the relative position of the spatial distribution pattern of elements, the main trend direction of development, and the degree of dispersion in the main and secondary directions. In this study, ArcGIS was used to generate the standard deviation ellipse of NPP in the study area to identify the position of the center and the spatial movement trend of NPP from 1997 to 2017.

### 2.3.7. Structural Equation Model

The structural equation model (SEM) [9,54–58] is an advanced and robust multivariate statistical method that combines factor analysis and regression analysis, allowing hypothesis testing on a complex network of path relationships to analyze the relationship between measured variables and latent variables, as well as the relationship between each latent variable. The SEM is composed of a measurement model and structural model. The former is used to analyze the relationship between measurement variables and latent variables, and the latter is used to analyze the relationship between latent variables.

The SEM can study not only observable variables, but also the relationship of variables that cannot be observed directly [59–61]. The SEM can be evaluated from many aspects, such as model regression coefficient, load coefficient, and model fitting index. In this study, a chi-square degrees of freedom ratio ($\chi^2/df$), comparative fit index (CFI), and root-mean-square error of approximation (RMSEA) were used to evaluate the model [62].

## 3. Results

### 3.1. NPP Estimation

The selected variables were brought into three models, including the RF, MLR, and BP neural network, to establish forest NPP remote sensing estimation models, which were validated using 10-fold cross-validation. The specific prediction accuracy evaluation is shown in Figure 2.

Comparing the prediction accuracy of the three estimation models, the $R^2$ of the RF (0.492–0.660) was higher than the MLR (0.307–0.532) and BP neural network (0.422–0.471) models in every year. RF sampling was performed twice. Firstly, the algorithm obtained a sampling set of training samples by random sampling with put-back. Then, a variable was randomly selected from all variables. Meanwhile, the best segmentation feature was selected as a node to build a classification and regression tree. The above reasons made the final model of RF have strong generalization and the highest prediction accuracy of NPP. Compared with traditional machine learning, the BP neural network requires more data to support. In this study, there were only 388 fixed sample plots in the study area, thus

the prediction accuracy of NPP was low. The study area has large elevation differences and complex terrain; therefore, the linear correlation between the dependent variable and most of the characteristic variables was poor, resulting in the MLR having the lowest prediction accuracy of NPP. The optimal RF model was selected for spatial mapping of forest NPP in the study area. The mean NPP values in 1997, 2002, 2007, 2012, and 2017 were 5.66 t·hm$^{-2}$·a$^{-1}$, 7.68 t·hm$^{-2}$·a$^{-1}$, 8.17 t·hm$^{-2}$·a$^{-1}$, 8.25 t·hm$^{-2}$·a$^{-1}$, and 10.52 t·hm$^{-2}$·a$^{-1}$, respectively. The mean and standard deviation of the forest NPP was calculated for the five periods and the NPP was classified into five classes, including low, medium-low, medium, medium-high, and high, by using the mean value of NPP minus 1-fold standard deviation, plus 1-fold standard deviation, and plus 2-fold standard deviation. The NPP mapping of each year is shown in Figure 3.

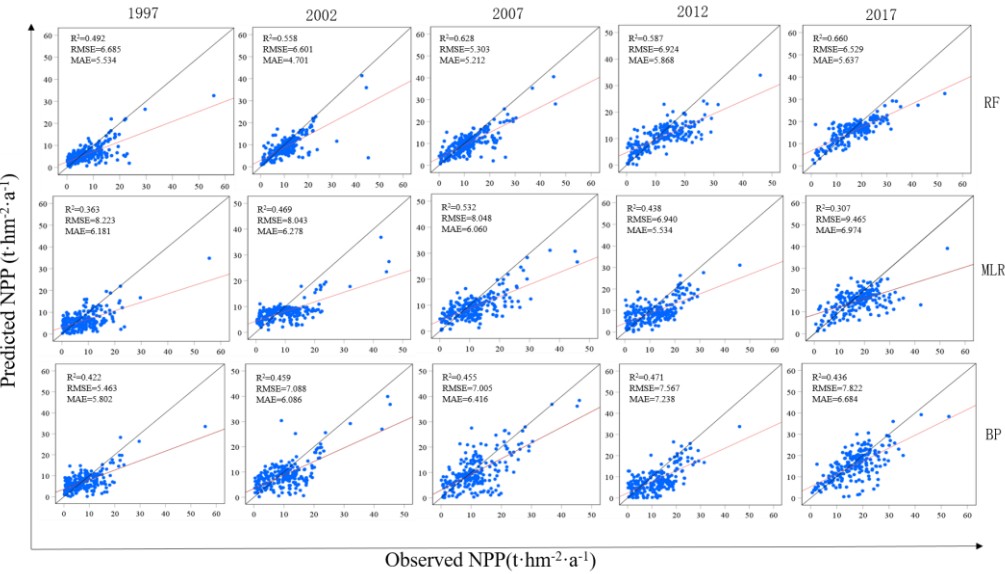

**Figure 2.** The performance of different inversion models in different years.

From the NPP grade classification mapping in 1997, 2002, 2007, 2012, and 2017, it can be seen that the distribution of high and low values of forest NPP was relatively consistent: the NPP of forests in the north, southwest, and west of the study area was higher, and the NPP of forests in the middle, northwest, and northeast was lower, which is relatively consistent with the altitude distribution of the study area. The northern, southwestern, and western parts of the study area have higher elevation, which are mostly mountainous, less densely populated, and less disturbed by human activities, and the forest vegetation can grow naturally, whereas the central, northwestern, and northeastern parts are mostly hilly areas and valley basins with lower elevation, which are densely populated and more urbanized, with low forest cover and more disturbance from human activities. It can be seen that the spatial distribution of forest NPP is highly consistent with the geomorphic characteristics and socio-economic conditions of the study area.

The forest NPP over 20 years, from 1997 to 2017, shows that the percent of low-grade NPP is gradually increasing, which is mainly due to the economic development and the expansion of the area of towns and cities. The area percentage of medium-low-grade NPP gradually decreased, as the forest with medium-low-grade NPP was farther away from the town compared to the low-grade NPP and was less affected by economic development and human activities. Meanwhile, with the implementation of forest land protection and management regulations, the forest of this grade was intensively managed and the forest NPP had been increasing, making the medium-low-grade NPP gradually develop to medium and medium-high grade. The area proportion of forest with medium-grade and high grade of NPP basically remained stable. The area proportion of medium-high-grade NPP is gradually increasing, as the forest of this grade is mainly located in the hills and river valley basin far away from the town. At the same time, with the enforcement of laws

for forest resources and forest land protection, the phenomenon of indiscriminate logging was obviously reduced, making the forest NPP gradually increase. It should be noticed that the area proportion of low-grade NPP in the study area has increased significantly due to urbanization development.

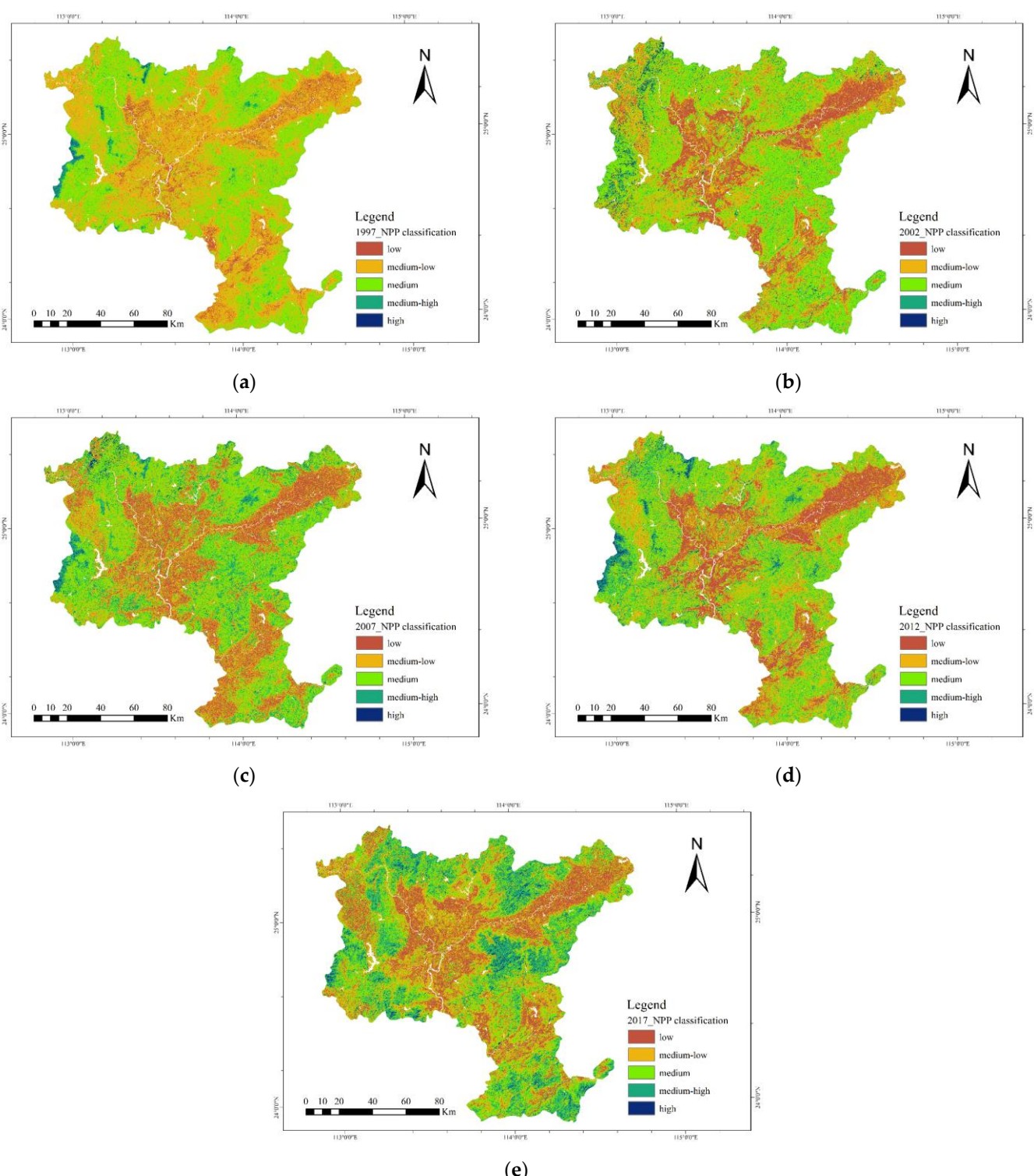

**Figure 3.** (**a**–**e**) are the spatial distribution maps of NPP classification in Shaoguan in 1997, 2002, 2007, 2012, and 2017, respectively.

*3.2. NPP Spatial and Temporal Dynamics*

3.2.1. Temporal Dynamics of NPP

Figure 4 shows the trend of the temporal dynamics of NPP in the study area from 1997 to 2017. Generally, the forest NPP in the study showed an upward trend from 1997 to 2017, with the increasing part being 69.86%, the non-significantly increasing part being 47.80%, the least-significant increase part being 7.37%, the significantly increasing part being 9.68%, and the extremely significantly increasing part being 5.02%. The areas of negative increase accounted for 30.12%, the areas of non-significant decrease accounted for 25.10%, the areas of least-significant decrease accounted for 1.59%, the areas of significant decrease accounted for 1.78%, and the areas of extremely significant decrease accounted for 1.67%. The areas with increasing NPP in the study area were mainly located in the mountainous and hilly areas in the west, southwest, and east-central parts of the study area; the areas with decreasing NPP were mainly located in the urban areas in the central, south-central, and northwest parts of the study area.

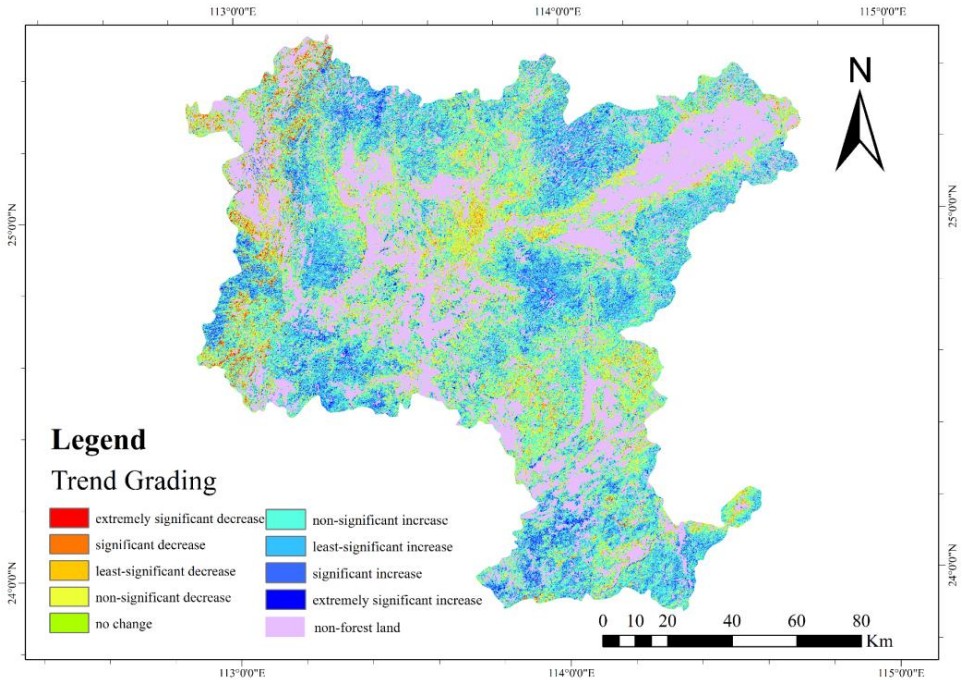

**Figure 4.** Temporal changes of NPP in Shaoguan from 1997 to 2017.

The NPP in the study area showed an upward trend during the 20 years from 1979 to 2017. In the areas close to the towns, there was a slight downward trend in NPP due to economic development and disturbance from human activities. In the areas far away from the towns, the active adjustment of stand species composition and stand structure made the overall NPP in the study area show an increasing trend.

3.2.2. Spatial Dynamics of NPP

Figure 5 shows the schematic diagram of the NPP standard deviation ellipse and the center point in the study area from 1997 to 2017. The short axis (i.e., *x*-axis) of the standard deviation ellipse of forest NPP from 1997 to 2002 became longer, the long axis (i.e., *y*-axis) became shorter, the flatness decreased, and the spatial aggregation increased, indicating that the forest NPP was gradually aggregated from the original uniform distribution and the distribution center shifted to the southeast. This is mainly due to the fact that around 1997, China entered a period of rapid economic development and economic overheating occurred, which led to an increase in forest logging in the low elevation areas of the study area. During the period of 1997–2002, economic rectification measures were gradually implemented, and the destruction of forest resources was slowly reduced. However, during

this period, the forest vegetation in the low elevation areas was not immediately restored, and the forest vegetation productivity remained high in the high elevation areas, causing an enlargement in the productivity gap between forest with low elevation and forest with high elevation and an increase in aggregation. From 2002 to 2012, the short axis of the standard deviation ellipse of forest NPP in the study area gradually became shorter, the long axis also gradually became shorter, the flatness decreased, and the spatial aggregation became shorter, indicating that the distribution of forest NPP in the study area was gradually uniform and the distribution center of forest NPP still shifted to the southwest. This is due to the vigorous development of regional ecological construction that focused on forestry development. More afforestation and greening efforts were made in economically developed, low-altitude areas, resulting in a gradual reduction of the gap with high-altitude, economically backward areas. From 2012 to 2017, the short axis of the standard deviation ellipse of forest NPP in the study area became longer, the long axis became shorter, and the flatness decreased, indicating that the spatial aggregation of forest NPP in the study area increased and the center shifted to the southeast. During this period, several forest-related policies were made and forest protection measures were strengthened in areas with a higher altitude and more backward economy; thus, forests were less subjected to human interference and trees grew vigorously, whereas low altitude areas were susceptible to the conversion from high NPP forest land to built-up economic areas. Therefore, the phenomenon of increased spatial aggregation of forest NPP occurred. Generally, from 1997 to 2012, the spatial aggregation of forest NPP in the study area increased and the NPP distribution center shifted from the central-western region to the southwestern direction. From the above analysis, it can be seen that the spatial variation of forest NPP in the study area was more influenced by forestry policies and socio-economic development, such as logging, and the conversion of forest land to built-up economic areas also had some negative effects on it.

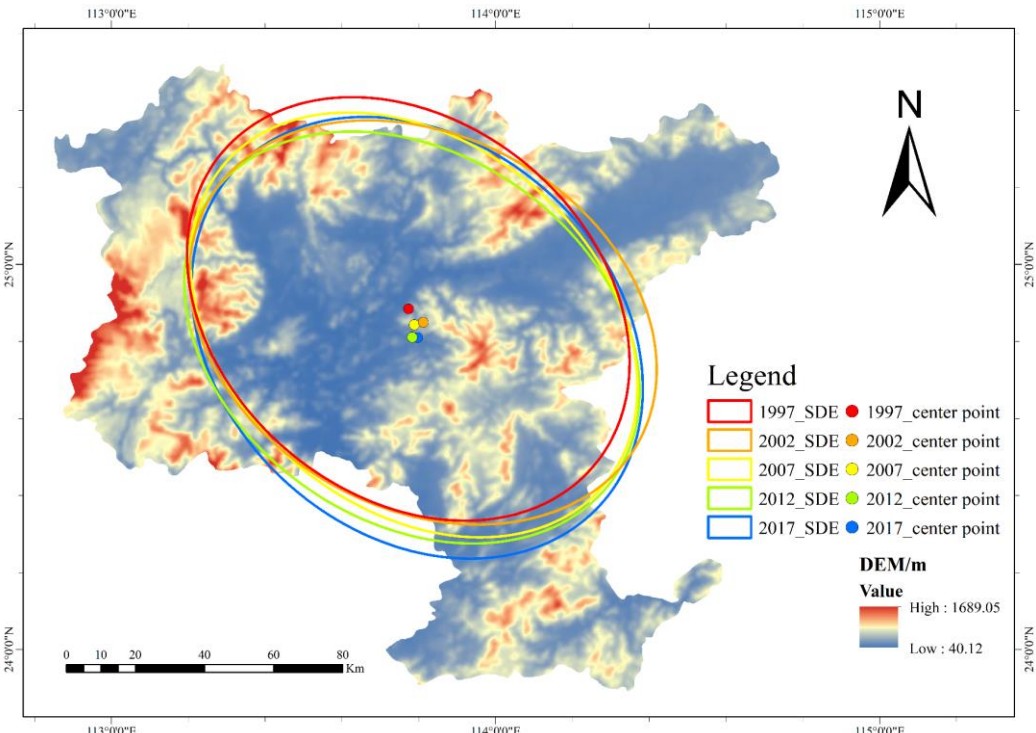

**Figure 5.** Spatial dynamic changes of NPP in Shaoguan from 1997 to 2017.

### 3.3. Driving Factors for NPP

Forest growth is influenced by environmental factors, such as topographic factors that can affect the distribution and allocation of light, precipitation, and forest soil, which affects forest productivity. Forest growth is also influenced by its own characteristics,

such as understory vegetation, forest stand structure, etc. There is a certain relationship between these factors which affect forest stand growth and forest NPP through interaction. In this study, three types of drivers including environmental factors, understory factors, and forest stand factors in 2017 were chosen to analyze their interactions and effects on forest NPP. Five factors, including slope direction, slope position, slope gradient, elevation, and landform, reflected topographic factors; whereas ten factors, including soil type, soil texture, soil thickness, humus thickness, litter thickness, shrub height, shrub cover, herb height, herb cover, and vegetation cover, reflected understory factors; and nine factors, including dominant species, species structure, age group, average age, average diameter at breast height (DBH), average tree height, canopy density, naturalness, and stand volume, reflected stand factors.

The driving factors were inputted into the SED model, and only 21 driving factors were retained after repeated tests; finally, the optimal SED model was obtained (Figure 6). $\chi^2/df$ was 1.99, GFI was 0.954, and RMSEA was 0.064, indicating that the fit of the constructed SED model was basically ideal.

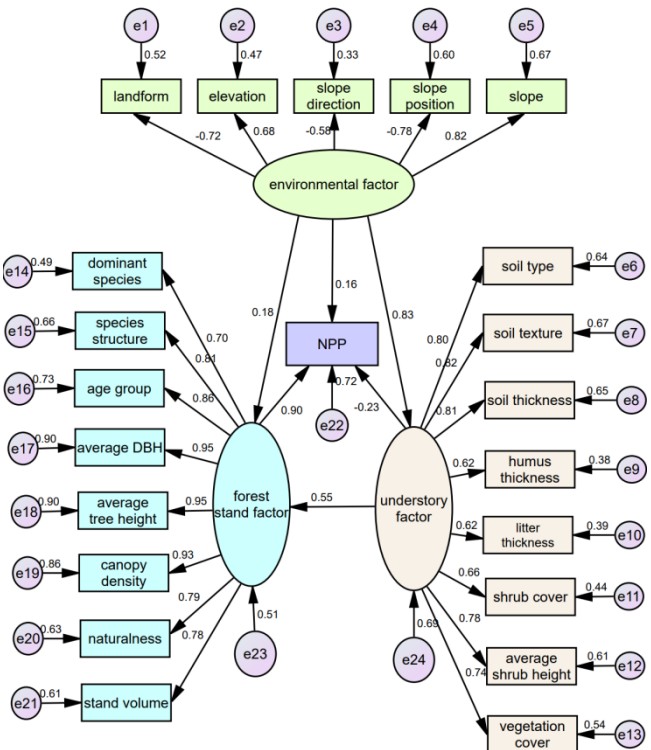

**Figure 6.** Schematic diagram of NPP structural equation model in Shaoguan in 2017.

As shown in Figure 6, the forest stand factor had the greatest influence on the forest NPP (the path coefficient of 0.90), followed by the understory factor which was negatively correlated with the forest NPP (the path coefficient of −0.23), whereas the environmental factor had the least influence (the path coefficient of 0.16), the environmental factor had a significant positive influence on both the forest stand factor and the understory factor with path coefficients of 0.18 and 0.83, respectively, and the understory factor had a significant positive influence on the forest stand factor with a path coefficient of 0.55. All the driving factors shown in Figure 6 reached significant levels. The standardized factor loading coefficients of environmental factors were slope (0.82), slope position (−0.78), landform (−0.72), elevation (0.68), and slope direction (−0.58), ranking in descending order. Among the forest stand factors, the highest standardized factor loading coefficients of 0.95 were for average DBH and average tree height, followed in descending order by canopy density (0.93), age group (0.86), species structure (0.81), naturalness (0.79), stand volume (0.78), and dominant species (0.70). The loading coefficients of standardized factors in the understory

were, from largest to smallest, soil texture (0.82), soil thickness (0.81), soil type (0.80), average shrub height (0.78), vegetation cover (0.74), shrub cover (0.66), humus thickness (0.62), and litter thickness (0.62).

The average DBH and the average tree height in a forest stand directly determine the forest biomass and, thus, the forest NPP; therefore, there is a strong positive correlation between forest stand factor and NPP. The environmental factors including the slope, slope direction, slope position, and elevation directly affected the absorption of light, heat, and nutrients by the forest; thus, the environmental factors have a positive influence on the forest NPP. The understory factors showed a negative relationship with the forest NPP, because the understory plants competed with arbor trees for nutrients and living space, and the more vigorous the understory plants, the slower the arbor trees grew, which eventually reduced the growth rate of NPP.

## 4. Discussion

In this study, three models, including the RF, MLR, and BP neural network, were applied to estimate forest NPP in the study area. After the selection of feature variables and valuation of the performance of the three models, the optimal model with the highest prediction accuracy was used to predict forest NPP. Based on the NPP prediction results of the optimal model, a spatial and temporal dynamic analysis and driver analysis were conducted to evaluate the long-term effects of forestry policies, human economic activities, and urbanization processes on forest NPP, thus providing some scientific basis for long-term sustainable forest management planning at the regional scale.

Among the three estimation models, the performance of the RF model was better than the MLR and BP neural network models. Using the optimal model of RF, the NPP estimation results were obtained in the study area. The average NPP in Shaoguan increased from 5.66 $t \cdot hm^{-2} \cdot a^{-1}$ in 1997 to 10.52 $t \cdot hm^{-2} \cdot a^{-1}$ in 2017. The average increase of NPP in the study area was 0.24 $t \cdot hm^{-2} \cdot a^{-1}$. An increasing trend was seen in the tropical zone, where an average increase of about 0.15 ($\pm 0.71$) $t \cdot hm^{-2} \cdot a^{-1}$ occurred. A change of more than 1.00 $t \cdot hm^{-2} \cdot a^{-1}$ was noted in the subtropical zone [63]. NPP by forest types were figured out using the sample plots data from 1997 to 2017 in the study area. The order of NPP from largest to smallest by forest types is: bamboo forest (21.90–28.77 $t \cdot hm^{-2} \cdot a^{-1}$), broad-leaved mixed forest (15.76–17.99 $t \cdot hm^{-2} \cdot a^{-1}$), broad-leaved forest (7.85–15.33 $t \cdot hm^{-2} \cdot a^{-1}$), mixed coniferous forest (11.82–15.07 $t \cdot hm^{-2} \cdot a^{-1}$), mixed coniferous forest (9.46-12.65 $t \cdot hm^{-2} \cdot a^{-1}$), coniferous forests (3.91–10.09 $t \cdot hm^{-2} \cdot a^{-1}$) and shrub forests (2.40–3.32 $t \cdot hm^{-2} \cdot a^{-1}$). Based on the NPP grading results, it can be seen that although the NPP of the study area increased during the 20 years from 1997 to 2017, the proportion of medium- and low-grade parts in the study area were higher. The Theil–Sen estimation and Mann–Kendall trend analysis showed that the area with increasing forest NPP in the Shaoguan area accounted for 69.86% and was mainly located in mountainous and hilly areas. The area with decreasing forest NPP accounted for 30.12% and was mainly in the built-up area of the town. The SDE showed that the spatial aggregation of NPP in the study area increased from 1997 to 2017, with the distribution center shifting to the southwest. The SEM showed that NPP was significantly positively correlated with forest stand factors and environmental factors, and negatively correlated with understory factors. The method summarized in this study, including the selection of feature variables, introduction of FCD, and spatio-temporal change analysis using Theil–Sen, Mann–Kendall, and SDE, can be applied to NPP estimation in other subtropical regions.

The average NPP in the same period was about 7.00 $t \cdot hm^{-2} \cdot a^{-1}$ [64] in southern China, 6.50 $\pm$ 3.00 $t \cdot hm^{-2} \cdot a^{-1}$ in Asia, and 5.89 $\pm$ 2.60 $t \cdot hm^{-2} \cdot a^{-1}$ in North America [65], which shows that the average NPP of the study area is higher than the above areas. The reason lies in the fact that the forests in the study area are dominated by young and middle-aged forests with faster growth rates. Besides, Shaoguan is a key forest area in China, with better forest land quality, higher forest management intensity, and more attention to forest protection. The mean value of NPP in South America was 9.21 $\pm$ 3.79 $t \cdot hm^{-2} \cdot a^{-1}$,

the mean value of NPP in the tropics was $10.78 \pm 3.40$ t·hm$^{-2}$·a$^{-1}$ [65], and the mean value of NPP in the subtropics was about 12.76 t·hm$^{-2}$·a$^{-1}$ [66], which shows that the mean NPP in the study area is lower than in these regions and even lower than in the subtropics, mainly due to the high number of young and middle-aged forests and small storage volume per unit area; thus, the age group structure should be actively adjusted to increase the proportion of mature and over-mature forests, whereas the rotation period of trees should be extended. We should also choose long-lived native broadleaf species and high carbon-fixing efficient species for planting. At the same time, more mixed forests of conifers and broadleaf trees should be established, and stand thinning measures be taken in young and middle-aged forests, so that the trees get enough light and nutrients to improve photosynthetic efficiency. The growth of understory vegetation should be reasonably controlled, and a certain thickness of litter on forest soil should remain to maintain soil productivity. The spatial and temporal dynamics of NPP in the study area are greatly influenced by forestry policies and socio-economic conditions. For example, China implemented the return of some forest land to local ownership in 1981, cessation of cutting in 1985, and removal of agricultural pursuits on forest land in 2003 [67]. Therefore, we should continue to strengthen the implementation of forestry policies, such as returning farmland to forests, constructing beautiful countryside, and compensating for ecological benefits of public welfare forests, in order to continuously reduce the adverse effects of socio-economic development and urbanization on forest productivity.

The extraction and screening of model feature variables, the modeling method, and the introduction of FCD variables can be applied to the study of forest NPP in subtropical regions. In this study, we used a linear model and machine learning methods to estimate NPP. In the future, deep learning models such as convolutional neural networks (CNN) should be introduced to compare their performance with linear models and machine learning methods. In addition to forest canopy density, the forest stand structure factors, including tree height, forest age, and other stand structure factors, can be added to the feature variables of estimation models to research whether they can improve the prediction accuracy of NPP.

## 5. Conclusions

In this study, the prediction accuracy of forest NPP using the RF model was better than other machine learning models and linear models. The introduction of forest canopy density (FCD) improved the NPP modeling accuracy. The NPP in the study area has gradually increased, but the tree species composition and age group structure still remain unreasonable. The spatial variation of forest NPP in the study area is more influenced by forestry policies, social development, and human disturbance. The NPP in the study area is significantly influenced by stand factors, followed by understory factors and environmental factors.

**Author Contributions:** Conceptualization, T.L. and M.L.; methodology, T.L.; software, T.L.; validation, T.L., M.L. and F.R.; formal analysis, T.L.; writing—original draft preparation, T.L.; writing—review and editing, F.R., M.L. and L.T.; visualization, T.L.; supervision, M.L.; project administration, M.L.; funding acquisition, M.L. All authors have read and agreed to the published version of the manuscript.

**Funding:** This research was funded by the National Natural Science Foundation of China, grant number 31770679.

**Conflicts of Interest:** The authors declare no conflict of interest.

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
