# Peer review of "Estimation and Spatio-Temporal Change Analysis of NPP in Subtropical Forests: A Case Study of Shaoguan, Guangdong, China"

_remotesensing, doi:10.3390/rs14112541_

Round 1

Reviewer 1 Report

This paper is methodologically well conceived. A significant contribution is made by the analysis of a large number of variables and the analysis for the twenty-year period of 1997. I think the paper lacks conclusions after the discussion.

Reviewer 2 Report

Exploring the spatial and temporal dynamic characteristics of regional forest net primary productivity (NPP) in the context of global climate change can not only provide a theoretical basis for terrestrial carbon cycle studies, but also provide data support for medium- and long-term sustainable management planning of regional forests. Therefore, the topic of paper is relevant.

The scientific novelty lies in the analysis of spatio-temporal dynamics and driving factors of forest NPP over long time based on Theil-Sen Median slope estimation, Mann-Kendall trend analysis, SDE and SEM for Shaoguan, Guangdong, China.

This study contributes with a fundamental tool to guide oak forest conservation and restoration efforts for fragments of all sizes and different spatial arrangements.

In the introduction, the relevance of the study is well substantiated and the state of the problem is described. The research objectives are formulated clearly and clearly.

The research was carried out in Shaoguan City which is located d in the northern part of Guangdong Province. The research area is described satisfactorily. The authors provided a visual map. The relief and forest vegetation of the study area is very diverse, which makes the study very interesting. Nevertheless, I would like to see a more detailed description of the forest vegetation. Ideally, if the authors provide a map of forest types in accordance with the ecological forest classification used in China. Or the authors can provide tables on the diversity and distribution of forest types with characteristics of the species diversity of various forest layers, spatial homogeneity and age structure. The authors analyze factors (age, species diversity, productivity of subordinate layers, and others), but do not give any characteristics for it in the paper. This makes it somewhat difficult to understand and may mislead readers. Therefore, it is desirable to describe the objects of research in more detail.

The methodology is described in detail. The authors used modern methods of analysis. The choice of methods is reasonable and adequate for the tasks set. Landsat images and National Forest Continues Inventory (NFCI) data in the corresponding years were used as the main data source. Three models of Random forest (RF), multiple linear regression (MLR), and BP neural network were applied to estimate forest NPP in the study area. Theil-Sen and Mann-Kendall trend analysis, standard deviation ellipse (SDE) were chosen to analyze its spatial and temporal dynamic characteristics, while structural equation modeling (SEM) was used to analyze the driving factors of NPP changes.

The research results are perfectly illustrated with figures and tables that are informative and do not duplicate each other.

The results show that the NPP in the study area has an increasing trend. The authors found that the forest stand factors have the greatest effect on NPP in the study area. It is also revealed that although forest NPP has fluctuated due to the change of forestry policies and human activities, forest NPP in Shaoguan has huge potential of increase in the future. The reliability of the results is beyond doubt, as a huge amount of data has been collected and analyzed based on modern analysis methods.

There is no decryption of "A" in Table 1. What does "age group" mean? What values can "A" take?

Also, the results would be more interesting and the conclusions more in-depth if the analysis were carried out by forest types. The mention of forest types is only in Table 1. Without an analysis of forest types, the paper may not be fully understandable. Since it is not clear what forests we are talking about. Moreover, the forest vegetation in the research area is very diverse.

Conclusions follow from the results and are reasonable.  The article will be of interest to a wide range of readers whose scientific interests are related to ecology, forestry, as well as climate change. Despite the fact that English is not my native language, I read the paper with interest and had no difficulties in understanding. The paper fully corresponds to the subject and level of the Remote Sensing.

However, the authors need to clearly formulate the theoretical and practical significance of their research, the scope of application, and the limitations of application. It would also be interesting if the authors point out unresolved issues and outline directions for further research.

Reviewer 3 Report

The abstract can be improved by providing more information on the findings especially the outcome of the comparison of models and the driving factors of NPP change. The statement that Shaoguan forest has huge potential of increase in the future is not substantiated in the paper and should be deleted. Instead, add a more plausible and useful conclusion to the abstract.

The methods section would benefit from having more background to the study area. Figure 1: Can a map of the forest types be included as reference? Also, please add more descriptive information on the type of forests in Shaoguan City for readers outside China. The exotic species such as Eucalyptus robusta – are they in streetscapes, parks or plantations? What is the status of these forests – are all now conservation forests? Also provide information on the key forest management policy changes that were implemented during the timeframe under investigation e.g. cessation of cutting, return of some forest land to local ownership, any removal of agricultural pursuits on forest land, etc.

State whether the SFA is responsible for establishing the inventory plots and gathering data.

State whether below-ground biomass data are included in the NFCI process.

What software process was used to repair the 2007 and 2012 image data sets?

The discussion is rather brief. How does the temporal trend of NPP compare with other tropical and subtropical forests? Given that the prediction accuracy of the best model was quite low for auditing purposes for carbon offsets/trading, how might the NPP classification be improved for these subtropical forests in the future? What further research should be undertaken?

The paper makes a very useful contribution to the field but requires checking for errors (e.g. National Forest Continues Inventory – do you mean continuous inventory? and some references are incomplete) and improvement in English expression, particularly in the Introduction and Discussion. I suspect that broad statement “China's subtropical forest is a unique forest ecosystems in the world at the same latitude, characterized by rich forest types, a wide range of tree species, and high forest productivity” is possibly untrue and in any case requires references to substantiate.

Round 2

Reviewer 2 Report

The authors have responded to all my comments. I like the revised paper. I believe that the paper fully corresponds to the subject and scientific level of Remote Sensing.